# Estimates of mean residence times of phosphorus in commonly-considered inorganic soil phosphorus pools

Julian Helfenstein[1,a]*, Chiara Pistocchi[2]*, Astrid Oberson[1], Federica Tamburini[1], Daniel S. Goll[3,b], Emmanuel Frossard[1]

[1]Institute of Agricultural Sciences, ETH Zurich, Lindau 8315, Switzerland
[2]Eco&Sols, Montpellier SupAgro, University of Montpellier, CIRAD, INRA, IRD, 34060 Montpellier, France
[3]Le Laboratoire des Sciences du Climat et de l'Environnement, IPSL-LSCE CEA/CNRS/UVSQ Saclay, Gif-sur-Yvette, France
[a]now: Agroscope, Zurich 8046, Switzerland
[b]now: Institute of Geography, University of Augsburg, Augsburg, Germany

*these authors contributed equally.

*Correspondence to*: julian.helfenstein@usys.ethz.ch, chiara.pistocchi@supagro.fr

**Abstract.** Quantification of turnover of inorganic soil phosphorus (P) pools is essential to improve our understanding of P cycling in soil-plant systems and improve representations of the P cycle in land surface models. Turnover can be quantified using mean residence time (MRT), however, to date there is little information on MRT of P in soil P pools. We introduce an approach to quantify MRT of P in sequentially-extracted inorganic soil P pools using data from isotope exchange kinetic experiments. Our analyses of 53 soil samples from the literature showed that MRT of labile P (resin- and bicarbonate extractable P) was on the order of minutes to hours for most soils, MRT in NaOH-extractable P was in the range of days to months, and MRT in HCl-extractable P was on the order of years to millennia. Multiple regression models were able to capture 54 – 63 % of the variability in MRT among samples, and showed that land use was the most important predictor of MRT of P in labile and NaOH pools. MRT of P in HCl-P was strongly dependent on pH, as high pH soils tended to have longer MRTs. This was interpreted to be related to the composition of HCl-P. Under high pH, HCl-P contains mostly apatite, with a low solubility, whereas under low pH conditions, HCl-P may contain more exchangeable P forms. These results suggest that current land surface models underestimate the dynamics of inorganic soil P pools and could be improved by reducing model MRTs of the labile and NaOH-P pools, considering soil-type dependent MRTs rather than universal exchange rates, and allowing for two-way exchange between HCl-P and the soil solution.

# 1 Introduction

Since only a small fraction (usually < 1%) of soil phosphorus (P) is present as phosphate in the dissolved state where it can be taken up by plants and microbes, the rate at which this pool is replenished from other soil P pools is critical to assess the bio-availability of soil P (Syers et al., 2008). The extent and the time scale on which unavailable soil P forms can become gradually bioavailable is an important factor affecting ecosystem productivity under increasing carbon dioxide concentrations (Sun et al., 2017). Estimates of P availability thus directly influence inferences on carbon sequestration. However, currently P availability is poorly constrained in land surface models, which hampers our ability to project future carbon sequestration rates (Goll et al., 2012). Modelling the rate of replenishment from different pools requires knowledge of the mean residence time of P in these pools. While there is growing information on concentrations of soil P pools in soils (Hou et al., 2018b), and how they react to drivers such as crop management, land use change or changing climate (Feng et al., 2016; Negassa and Leinweber, 2009; von Sperber et al., 2017), little is known about the temporal dynamics of these pools (Hou et al., 2019). This knowledge gap not only limits linking data on soil P pools to P availability, it also slows the incorporation of P into global land surface models, and hence estimating the effect of P cycling on long-term ecosystem functioning at large spatial and temporal scales (Reed et al., 2015).

Soil P pools are most commonly measured using sequential extraction, whereby soil P is extracted with increasingly strong reagents to yield experimentally-defined P pools. While different variants of sequential extraction exist, the Hedley extraction and variants thereof are most widespread (Hedley et al., 1982). Here, we analyzed studies that used a resin-extraction, followed by 0.5 M $NaHCO_3$, 0.1 M NaOH, and finally 1 M HCl extractions. It is assumed that resin and $NaHCO_3$-extractable P represents loosely-sorbed P, NaOH extracts P associated to Fe and Al (hydr)oxides, and HCl-P contains phosphates associated to Ca and soluble in acids, especially apatites (Moir and Tiessen, 1993). A precise characterization of P mineral forms present in these inorganic pools is difficult since there is a plethora of mixed compounds and not pure crystalline P forms in soils. However, spectroscopic techniques have been used to confirm that soils with more HCl-P tend to contain more Ca-P, and soils with relatively larger NaOH pools P contain more P associated to Fe- and Al (Frossard et al., 2002; Helfenstein et al., 2018a; Kar et al., 2011; Prietzel et al., 2016; Wu et al., 2014).

Mean residence time (MRT) of P has been quantified in the soil solution and in soil microbes; however, little is known about the mean residence time of P in Hedley pools. Mean residence time is defined here as the average time required to completely renew the content of a pool at steady state, and is also called turnover time in other studies (Six and Jastrow, 2002). Radioisotopic labeling-experiments have shown that the MRT of P in soil solution is on the order of milliseconds to minutes (Fardeau et al., 1991; Helfenstein et al., 2018b), while MRT of P in soil microbiota tends to be on the order of days to weeks (Gross and Angert, 2017; Oberson and Joner, 2005; Spohn and Widdig, 2017). Isotope exchange kinetic

experiments, where the dilution of a radiosiotopic tracer ($^{33}$P or $^{32}$P) is traced in a soil-water suspension, allow measuring the exchange of P by physicochemical processes as a function of time (Fardeau, 1996; Frossard et al., 2011). While the MRT of P in resin, NaOH, HCl pools has recently been approximated in soils on a Hawaiian climatic gradient (Helfenstein et al., 2018a), it is not clear if these MRTs also pertain to non-volcanic soils.

Land surface models describe the complex interactions governing the cycles of water, energy, carbon, and increasingly incorporate cycles of major soil nutrients (i.e. nitrogen and phosphorus) (e.g. Wang et al. 2010). The representations of inorganic soil phosphorus dynamics are currently very rudimentary in such models: soil inorganic phosphorus is commonly separated into different pools according to differences in residence times (fast, intermediate and slow pools) (Wang et al., 2010). Although the structure, i.e. number of soil P pools and the connection among pools, differs among

models, they commonly apply the same concept. In general, the residence time of P in the fastest pool is modelled as a function of multiple abiotic (chemical weathering, sorption) and biotic (mineralization, immobilization, plant uptake, phosphatase activity) processes (Lloyd et al., 2001), while more recalcitrant pools have prescribed globally uniform decay rates. The decay rates are either derived from calibration to achieve plausible total soil P stocks (Goll et al., 2012; Wang et al., 2010; Yang et al., 2014), or derived from temporal net changes in soil P fraction along a single soil chronosequence (Goll et al., 2017). Such calibration strategies are not able to capture gross exchanges among soil P pools, which do not result in

net changes in pools size, but are critical for the bioavailability of soil P. Hence, current model formulations of P cycling are likely underestimating inorganic soil P dynamics and overestimating inorganic P pool MRTs.

The objective of this work was to quantify the MRT of P in inorganic soil P pools in a wide range of soil types and provide much needed information to constrain soil P dynamics in global models. We limited our study to inorganic pools because the isotope exchange kinetic approach can only be

used to study physicochemical exchange processes, whereas MRT of P in organic pools is controlled by biological processes. In our analysis, we assume that the time necessary to renew the total amount of P present in a Hedley pool (MRT) can be equated to the time necessary for phosphate ions in solution to exchange with all phosphate ions located in this pool (Fig 1a). This allowed us to calculate MRT using the function for isotopically exchangeable P as determined in isotope exchange kinetic experiments. However, it also means that potentially important processes influencing P MRT, such as biological and solid-state P transformations, were neglected. The second objective of this work was to determine if variation in MRT

among soils could be explained by soil properties and climatic variables. To meet these aims, we searched the literature for soil samples for which both P sequential extraction and IEK data were available. We then used a function describing isotopically exchangeable P as a function of time to calculate MRT of P in nesin-P, labile-P (resin- and bicarbonate extractable), NaOH-P, and HCl-P.

## 2 Methods

### 2.1 Data set

We found 53 soils for which both P sequential extraction and isotope exchange kinetic data were available. Twelve samples were from a Hawaiian climatic gradient (Helfenstein et al., 2018a), five from a long-term field trial in Switzerland (Keller et al., 2012), eleven from a study of different soils throughout New Zealand (Chen et al., 2003), fourteen from a forested geosequence in Germany (Lang et al., 2017), seven from field-experiments on highly-weathered soil in Colombia (Buehler et al., 2003; Oberson et al., 1999), and four from an agricultural field-trial on Fluvisols in Italy (Borda et al., 2014). Information on soil properties (pH, soil texture, organic C concentration), climate (mean annual temperature and mean annual precipitation), soil P pools (total P, resin-extractable P, $NaHCO_3$-extractable inorganic P, NaOH-extractable inorganic P, HCl-extractable P), and P exchange properties ($P_w$, m, n, $P_{inorg}$, see section 2.2) were retrieved from the original publications, associated publications, or by contacting the authors (Table 1). One study used an ammonium chloride extraction instead of a resin-extraction (Chen et al., 2003). We added the first two extractions (resin or ammonium-chloride and $NaHCO_3$) and called this the "labile pool". For two studies (Borda et al., 2014; Helfenstein et al., 2018a) soil texture was not reported and had to be estimated from global gridded (250m resolution) soil information based (Hengl et al., 2017) based on geographic information of the sample site. For the full documentation of sources for the soil property information, please see Supplementary Table 1. Four soils ("Himatangi", "Hurundi", "Okarito", and "Temuka" (Chen et al., 2003)) were excluded because, unlike the rest of the samples, the Hedley pools of these soils were much larger than the pools from isotope exchange kinetics. This yielded unreasonably high MRTs because of the asymptotic behavior of the E(t) equation. In the discussion, we briefly discuss how excluding these soils affected multiple regression models.

Despite only encompassing 53 soils from 20 geographic sites, the dataset included samples from a wide set of climatic conditions (Supplementary Fig. 1), and a variety of soil texture classes (Supplementary Fig. 2). Eleven of the soils samples were from arable land use, 14 from forest, and 28 from grassland. The world reference base soil orders entailed were Acrisol (1), Andosols (17), Cambisols (19), Ferralsols (7), Fluvisols (4), and Luvisols (5) (IUSS Working Group WRB, 2015). While some of the soils are considered to be low in available P (Buehler et al., 2003; Lang et al., 2017; Oberson et al., 1999), the data set also includes soils developed on P rich parent materials (Helfenstein et al., 2018a; Lang et al., 2017) or soils that have been intensively fertilized (Borda et al., 2014).

## 2.2 Estimating mean residence times

Isotopically exchangeable P ($E_{(t)}$) [mg kg-1] can be modelled as a function of time by Eq. 1 (Fardeau, 1996), where $m$ and $n$ are statistical parameters, $P_w$ [mg kg-1] is water-extractable P, r(∞) is the radioactivity measured in solution after an infinite time and R is the radioactivity [Bq] introduced at t=0.

$$E_{(t)} = P_w * \frac{1}{m*\left(t+m^{\frac{1}{n}}\right)^{-n} + \frac{r_{(\infty)}}{R}} \tag{1}$$

The ratio $\frac{r_{(\infty)}}{R}$ is usually approximated as $\frac{P_w}{P_{inorg}}$, where $P_{inorg}$ is the total amount of inorganic P [mg kg-1] (Fardeau, 1993). The parameters $m$ and $n$ describe the rapid and slow physicochemical exchange processes, respectively, and are determined by fitting a non-linear regression model to measurements of radioisotope concentration in solution from an isotope exchange kinetic experiment (for details see Fardeau et al. 1991 and Frossard et al. 2011).

By plugging in different values of $t$ [min], one can calculate the amount of P that is exchangeable within a given time frame. Likewise, it is possible to calculate the amount of P exchangeable between two time points, e.g. between one day and 3 months (Frossard et al., 2011). While isotope exchange kinetic experiments used to fit $m$ and $n$ only last for around 90 minutes, extrapolated E-values have been shown to describe P exchange well over a time span of months, accurately describing P available to plants and being considered the gold-standard for measuring P availability (Frossard et al., 1994; Hamon et al., 2002). Furthermore, $E_{(1\ min)}$ has been shown to correlate with resin-P, $E_{(3\ months)}$ with NaOH-P and $E_{(>\ 3months)}$ with HCl-P in sewage sludge (Frossard et al., 1996) and for soils from a Hawaiian climatic gradient (Helfenstein et al., 2018a).

To estimate mean residence times of sequential extraction pools, we plugged in P pool values as measured by sequential extraction ($P_{pool}$) for $E_{(t)}$ (Eq. 2), and then solved for $t$ to determine the amount of time necessary to exchange all the phosphate ions contained in that pool (Eq. 3).

$$P_{pool} = P_w * \frac{1}{m*\left(t+m^{\frac{1}{n}}\right)^{-n} + \frac{P_w}{P_{inorg}}} \tag{2}$$

$$t = \left(\frac{\frac{P_w}{P_{pool}} - \frac{P_w}{P_{inorg}}}{m}\right)^{-\frac{1}{n}} - m^{\frac{1}{n}} = MRT \tag{3}$$

In sequential extractions, P pools are sequentially removed from the soil, and this has to be accounted in the calculation of MRT. MRT of resin-P and labile-P was calculated using Eq. (3) and plugging in resin-P or labile-P pool sizes for $P_{pool}$. However, for NaOH-P and HCl-P the previously

removed P pools have to be formally "added back". Hence, for NaOH-P and HCl-P the $P_{pool}$ was set equal to the sum of labile-P and NaOH-P or labile-P, NaOH-P, and HCl-P respectively (Fig. 1b). Not accounting for the sequential nature of these pools and using NaOH-P or HCl-P for $P_{pool}$ directly in Eq. (3) would underestimate MRT.

Estimating MRT using Eq. (3) required making several assumptions. Firstly, we assumed that the labile pool exchanges much faster than NaOH-P pool, which again exchanges much faster than HCl-P. This assumption rests on the observation that radioisotope specific activity after labeling is higher in resin-P >> NaOH-P >> HCl-P in a variety of soils (Buehler et al., 2002; Bünemann et al., 2004; Daroub et al., 2000; Pistocchi et al., 2018; Vu et al., 2010). Secondly, we assumed that all P transformations occur via the soil solution, i.e. we neglected potential exchange between pools in the solid phase (such as diffusive penetration, Fig 1a) (Barrow and Debnath, 2014). For example, we did not consider exchange between NaOH-P and "occluded" soil P forms, considered in some modelling approaches (Hou et al., 2019), but which do not involve phosphate release to the soil solution.

Thirdly, our approach carries over all assumptions from an isotope exchange kinetic experiment (Frossard et al., 2011), including that biological activity does not markedly impact P exchange during the duration of an isotope exchange kinetic experiment (usually 90 minutes). The assumption that biological activity is negligible at this time scale can be tested by applying a microbial inhibitor to the soil suspension (Bünemann et al., 2012). If the microbial inhibitor does not significantly affect radioisotopic recovery compared to the control, biological processes have a negligible effect. Microbial inhibitors did not affect radioisotopic recovery of soils from the Hawaiian climatic gradient or soils from the forested geosequence (only extreme soils tested, including a very low P soil) used in this study (Bünemann et al., 2016; Helfenstein et al., 2018a). However, for the other studies analyzed here, no such tests were undertaken. Hence, our approach can only be used to estimate MRT of P in inorganic P pools turning over by physicochemical processes, but does not account for biological processes. In our analysis, we also do not consider the residual P pool (P remaining in soil after the HCl-extraction), assuming that this pool only plays a negligible role in P exchange.

## 2.3 Effect of soil properties on mean residence times

To determine the effect of soil properties on MRT we performed multiple regression, in which independent soil properties were the explanatory variables and MRTs the response variables. Multiple regression models were fitted to the three response variables "t_labile", "t_NaOH", and "t_HCl". For each response variable, we derived a maximum scope model including all numerical soil properties (pH, soil texture, and organic C) and climatic data as well as categorical explanatory variables "land use" and "extraction method". Different studies used slightly different extraction methods, the effect of this on the variability was explored using the "extraction method" variable. Additionally, we tested the correlation of MRTs

with oxalate or dithionite-extractable aluminum and iron as a simple regression for the samples (n=41-43) for which this data was available. The R function "step" (R Core Team, 2018) was then used for stepwise selection of explanatory variables by minimizing Akaike's information criterion (AIC) (Sakamoto et al., 1986). Model size was reduced to reduce collinearity between predictor variables, as assessed using the Variance Inflation Factor (VIF), which was below 5 for each of the explanatory variable (Fox and Monette, 1992). Non-normally-distributed variables were log-transformed to meet the assumption of normality.

## 3. Results

### 3.1 P exchange as a function of time

P pools as defined by sequential extraction displayed highly significant correlations with P pools defined by exchangeability, with most points falling close to the 1:1 line (Fig. 2). Pearson's moment correlation between labile P and P exchangeable within 1 hour was 0.84, between NaOH-P and P exchangeable between 1 hour and 3 months was 0.94, and between HCl-P and P only exchangeable in time spans longer than 3 months was 0.87.

The soils showed a broad range of P exchange as a function of time. P that was exchangeable within 1 min ($E_{1min}$) ranged from 0.99 – 218 mg kg$_{-1}$, and P that was exchangeable in three months from 11.7 to 6311 mg kg$_{-1}$ between the different soils (Table 1). Soils developed on P-rich basalt (Helfenstein et al., 2018a; Lang et al., 2017) had the highest E-values; while Ferralsols had the lowest E-values (Oberson et al., 1999). Half of the soils had < 5 mg P kg$_{-1}$ exchangeable within one minute, which is considered a threshold for low P availability (Gallet et al., 2003). Differences in P exchange behavior were either due to different levels of total inorganic P, or to different P forms present in the soil (Fig 3). For example, a soil with high amounts of inorganic P exchanged more within the same time interval than a soil with low amounts of inorganic P. Similarly, soils with large proportions of HCl-P tended to have lower slopes of E-curves than soils with relatively more labile or NaOH-P. This variability is reflected in the proportion of $E_{1min}$ to total P, which spanned from 0.04 to 6% of total soil P. Similarly, $E_{3months}$ represented on average 25% of total soil P, spanning from 4 to 64%.

### 3.2 Estimates of mean residence times

The median MRT of P in the labile pool was around 1 hour (67 min), for NaOH-P around 1 month ($3.4 \times 10^4$ min), and for HCl-P around three years ($1.4 \times 10^6$ min). However, calculated MRTs of individual soils spanned many orders of magnitude. Mean residence time of P in labile P

ranged from 0.4 to 4.4 x $10^3$ min, with two frequency maxima, one around one hour and one around one day (Fig. 4). Mean residence time of P in NaOH-P ranged from 91 to 3.4 x $10^6$ min, and also had two frequency maxima, one at around 1 day and one at around 3 months. Mean residence time of P in HCl-P had the widest spread, from 129 to 1.7 x $10^{15}$ min. While most soils had a MRT of P in HCl-P around 1 year, another frequency maximum occurred at around 10'000 years. The median MRT of P in resin P (n = 42) was 19 minutes, with a range of $10^{-4}$ to $10^2$ min.

**3.3 Soil properties affecting mean residence times**

Multiple regression models were able to explain between 54 and 63% of the variability in MRT for each pool (Table 2). The MRT of P in labile P was best predicted by a model including clay and land use (adj. $R^2 = 0.63$, $F$-statistic = 30.6, $p < 0.001$). Land use was the most important predictor of t_labile. A model only including land use had an adj. $R^2$ of 0.52 ($F$-statistic 29.6, $p < 0.001$). While most forest and grassland soils had a MRT of P in labile P around 1 h, arable soils tended to have a longer MRT of around 1 day (Fig. 5a). The land use effect on MRT of P in labile P was not a pool size effect (Fig. 5b). In addition, MRT decreased with increasing clay content (adj. $R^2 = 0.12$, $F$-statistic = 6.8, $p = 0.01$) (Supplementary Fig. 3).

The MRT of P in NaOH-P was best predicted by a model with clay, land use and organic C (adj. $R^2 = 0.57$, $F$-statistic = 18.4, $p < 0.001$). Of those three explanatory variables, the relationship was again strongest with land use. A model only including land use had an adj. $R^2$ of 0.44 ($F$-statistic 21.6, $p < 0.001$). In general, forest soils had a shorter MRT of around 1 day and arable soils a longer MRT of around 3 months. Grassland soils spanned the full range in MRT of P in NaOH-P (Fig. 5c). As for MRT in labile P, the land use effect on MRT was not a pool size effect (Fig. 5d). MRT of P in NaOH-P also decreased with increasing clay content (adj. $R^2 = 0.13$, $F$-statistic = 7.1, $p < 0.01$) (Supplementary Fig. 3). While organic C was also a significant predictor in the multiple regression model, a simple regression between organic C and MRT of NaOH-P was not significant (Supplementary Fig. 4).

The MRT of P in HCl-P was best predicted by a model with clay, pH, and mean annual rainfall (adj. $R^2 = 0.54$, $F$-statistic = 17.8, $p < 0.001$). The strongest of these predictors was pH (Fig. 6). MRT of P in HCl-P increased with increasing pH following Eq. (4):

$$log\ (t_{HCl}) = -7.95 + 4.63 * (pH) \tag{4}$$

where t_HCl is in minutes (adj. $R_2 = 0.47$, *F*-statistic = 37.7, p < 0.001). Like with the models for labile P and NaOH-P, predicted MRT decreased with increasing clay concentration also for MRT of HCl-P; however, this relationship was not significant as a simple regression (Supplementary Fig. 3).

### 4.1 Discussion

Sequential extraction is probably the most common method used to study P pool distribution in soils. However, the residence time of P in these pools and environmental controls remain poorly understood. While earlier works hypothesized that resin and bicarbonate P have a "fast turnover", and NaOH and HCl a "slow turnover", data on MRT of P in these pools for a wide range of soils was previously missing (Cross and Schlesinger, 1995; Tiessen et al., 1984). We found that on average resin-P has a MRT on a range of several minutes, labile of one hour (forest and grassland soils) or one day (arable soils), NaOH-P on a range of days (forest and some grassland soils) to months (arable soils), and HCl-P of years to

millennia, with a strong pH dependence. The large variability in MRTs could be partially explained by soil properties, especially pH and clay, or land use, but may also be due to unaccounted soil properties as well as methodological limitations of either our approach or the lab techniques used to produce the original data. For instance, some variability in the MRTs estimation might be generated by the different methods used to measure total inorganic P. The accuracy in total inorganic P measurement might affect MRTs as results from Eq. (2).

As a predictor of MRTs of labile and NaOH-P, land use is likely a proxy for soil P balance (fertilizer inputs, outputs with harvest) and biological activity. Arable soils are more likely to receive P fertilizers. Long-term fertilizers inputs may lead to a decrease in surface charge resulting from diffusive penetration of P into the reacting materials, and therefore to a lower phosphate buffering capacity (Barrow and Debnath, 2014). Hence, fertilizer application may lead to larger P pool sizes but longer MRT (Helfenstein et al., 2018b). Biological activity has been shown to accelerate P turnover in the labile pool through the rapid uptake and release by the soil microbial community (Oehl et al., 2001; Pistocchi et al., 2018; Weiner

et al., 2018). This holds especially true under grassland or forest and under low P availability. We consider the later explanation less likely, since microbial uptake/release tends to be negligible during the isotopic exchange kinetic experiments (Oehl et al., 2001) or it is suppressed using microbial inhibitors (Bünemann et al., 2012). However, we cannot completely rule out such an influence as in most soil samples included in our data-set this effect was not systematically tested.

The pH dependence of MRT in HCl-P is likely because the composition of the HCl pool varies strongly with pH. Under high pH, the HCl pool tends to contain apatites, Ca-P minerals which are highly stable (Moir and Tiessen, 1993; Nriagu, 1976). Our results predicts that under such

conditions, MRT of HCl-P may be on the order of millennia or longer, orders of magnitude longer than estimated MRTs of NaOH-P. In acidic soils on the other hand, apatite is much less stable (Guidry and Mackenzie, 2003), and the HCl pool may contain either carry-over from the NaOH pool or other phosphates that are more reactive (Prietzel et al., 2016). Eq. (4) predicts a MRT in HCl-P of ¾ year for a soil with pH 4.5, a range into which also MRT of many NaOH-P pools falls. Hence, our results suggest that the exchange kinetics of NaOH and HCl pools are more similar under low pH conditions, whereas under high pH conditions, there seems to be a clear distinction between availability of NaOH-extractable P and HCl-extractable P. Nevertheless, even under neutral to alkaline conditions, HCl-P pool may be involved in bi-directional exchange of phosphate involving precipitation of phosphate with Ca to form secondary Ca-phosphates, without net change in pool size (Frossard et al., 1995), although at pedogenetic time scales this pool progressively decreases in size (Walker and Syers, 1976). Also, using stable oxygen isotopes in phosphate, it was shown that in 150'000 year old soils from arid conditions and neutral pH, roughly half of the HCl-P pool contained secondary phosphates (Helfenstein et al., 2018a). The relationship of pH and other relationships, e.g. land use as a predictor for t_labile and t_NaOH, pertained if the outliers "Himatangi", "Hurundi", "Okarito", and "Temuka" were included in the multiple regression analyses; however, including the outliers reduced overall model adj. $R_2$ to around 0.4 for all three models.

Clays are important binding sites for P (Gérard, 2016). Our data seem to show that the clay content influences residence times not only of labile-P, but also of the NaOH and HCl pools. In our analysis, the clay variable includes not only clay minerals but also secondary minerals such Fe and Al oxyhydroxides, as it follows the particles size classification. Fe and Al oxyhydroxides are known to be key in inorganic P exchange behavior (Achat et al., 2016; Syers et al., 2008). For the samples where data on oxalate- and dithionite-extractable Al and Fe was available (n = 41-43), simple regression showed only weak correlations with MRTs, and only significance for MRT of NaOH-P with oxalate-extractable Al, dithionite-extractable Fe and the sum of dithionite-extractable Al and Fe as explanatory variables (adj. $R_2 \leq 0.16$, $p < 0.05$, data not shown). In general, soil properties controlling P sorption also control P exchangeability and therefore residence time. Indeed, high amounts of P-sorbents might relate to more rapidly exchangeable P (Achat et al., 2016; Demaria et al., 2013). The variety of mechanisms involved in P binding on such soil surfaces (multi-layer sorption, inner-sphere complexes, surface precipitation, see Gérard 2016 and references therein) might explain why the effect of clay is significant for all residence times.

Data from long-term (weeks-months) radioisotope tracer incubation experiments, where both physicochemical and biological processes are considered, support our estimates of MRTs. While such studies have not reported estimates of MRTs, the time by which specific activity of $_{33}$P of the NaOH or HCl pool equilibrates with the specific activity of the labile P pool provides an estimate of MRT of the slowest pool, i.e. the time needed to exchange all the phosphate ions located in the slowest pool with the ones in the soil solution. According to this assumption and using the

data published by Buehler et al. (2002) from the same Ferralsols also included in our dataset, we could estimate a MRT of the NaOH pool between 7 and 14 days (soils under savanna and pasture) or longer than two weeks (two soils under rice). These values are similar to MRTs from our study: 1 and 5 days for the soils under savanna and pasture, respectively, and 28 and 88 days for the soils under rice. Generally, specific activity of $^{33}P$ in HCl pool did not equilibrate during the duration of the experiment (two weeks to three months, depending on the study), suggesting longer MRTs

for this pool (Buehler et al., 2002; Bünemann et al., 2004; Pistocchi et al., 2018; Vu et al., 2010). Nevertheless, for stable pools such as the HCl-P, it is questionable whether our estimates of MRTs are realistic, as the extrapolation of E-values (Eq. 1) has been tested only over time spans of days to months (Bünemann et al., 2007; Frossard et al., 1994) and is impossible to validate for longer time spans due to the short half-lives of both radioactive P isotopes.

Insights from stable oxygen isotope analysis support our estimates of MRT of HCl-P. At the beginning of soil development, all soil P has the parent material stable oxygen isotope value ($\delta_{18}O_P$) (Roberts et al., 2015; Tamburini et al., 2012). With time, biological activity brings $\delta_{18}O_P$ into steady-state with soil water (Blake et al., 2005). By analyzing $\delta_{18}O_P$ in sequentially-extracted pools in soils of known age, it is possible to roughly constrain MRT of P in these pools. While $\delta_{18}O_P$ of bicarbonate- and NaOH-extractable P tend to be in the soil-water steady-state (Helfenstein et al., 2018a; Roberts et al., 2015), HCl-P may retain parent material signature even in older soils. In a chronosequence on granitic parent material, it was shown

that the HCl pool acquired the biological signature after several thousand years (Tamburini et al., 2012), whereas under more arid conditions, where apatite remains stable, HCl-P may not have turned over completely even after 150'000 years of soil development (Helfenstein et al., 2018a). This supports not only our long and variable estimates of MRTs of P in the HCl-P, but also their strong dependence on pH, the main driver of apatite stability.

**4.2 Limitations**

The main limitations of our study concern representativity of the soil samples used and uncertainty introduced due to assumptions taken to calculate MRTs. The 53 soils samples used in our study only came from a small number of studies, and some soils, like Andosols and Cambisols, were overrepresented in our study, while other important soils such as Vertisols, Podzols, or carbonate or organic matter-rich soil orders were not/under represented. In addition, soils with large amounts of NaOH- and HCl-P were overrepresented in our study compared to a larger global data set

(Supplementary Fig. 5). However, our resin P values closely match resin P frequency distribution of a larger global dataset (Hou et al., 2018a). In terms of P exchange kinetics, our soils covered the full range of reported m and n values, and can thus be considered to reflect the full range of P exchange kinetic properties observed in soils (Supplementary Fig. 6) (Achat et al., 2016; Helfenstein et al., 2018b). Assumptions taken to calculate

MRT of P in soil P pools required making several simplifications. Our approach only considers a simplified system of soil and water in steady-state conditions, and excludes biological activity. In field conditions, P residence times may be different due to non-steady state conditions, microbial interactions with abiotic processes, as well as plant uptake and alterations of the physical and chemical soil environment (Hinsinger, 2001). For example, intensive P uptake by plants may lead to net changes in soil P pools in addition to exchange fluxes (Guo et al., 2000). Also, it is likely

that MRTs are affected by temperature and changes in soil moisture. Continued improvement of tracer experiments is paramount to provide empirical data on mean residence times and magnitudes of biological and physicochemical fluxes (Bünemann, 2015; Wanek et al., 2019). However, for the time being, our *ad hoc* approach provides preliminary estimates of mean residence times of commonly used P pools, with the potential to improve both interpretation of lab-and field scale results as well as land surface modelling.

### 4.3 Implications for lab- and field-scale research

Mean residence times of P in inorganic soil P pools reported here are important for improved understanding of P dynamics in soil. Sequential extraction continues to be widely used to measure soil P status, for example to study effects of land use change and P inputs or environmental drivers on soil P cycling (Blake et al., 2003; Feng et al., 2016; von Sperber et al., 2017). Understanding the residence times of soil P pools will allow fine-tuning hypotheses of expected changes and improve interpretation of observed changes in pool sizes. Furthermore, analysis of stable oxygen isotopes in phosphate, which is gaining importance as a tracer of phosphate transformation and indicator of biological vs. geochemical P

cycling, is tightly linked to sequential extraction (Tamburini et al., 2018). Knowledge of mean residence times has the potential to improve interpretation of sequential extractions and derivate methods.

### 4.4 Implications for land surface modelling

Our study allows drawing several conclusions important for land surface modelling. Firstly, we show that current LSM largely overestimate MRTs of P in inorganic soil P pools. As models do not report all information needed to calculate MRT, we approximate modelled P pool MRT (in the

following: MRT*) of the intermediate and slow pool as the inverse of the decay rate assuming the pool size and fluxes are in equilibrium (i.e. net losses = net gains). Juxtaposing our estimates of MRTs of P in inorganic soil P pools with values used in existing land surface models shows that existing land surface models underestimate inorganic P turnover by several orders of magnitude (Table 3). While our estimates of the mean residence times of the NaOH pool are on the range of months to years, existing models run with MRT*s of these pools of decades to centuries (Table 3). The discrepancy between our estimates and existing model values is so large and consistent among models that it is unlikely due to slightly different

conceptualizations of the P cycle or the approximation of MRT in models by decay rates. Rather, the overestimation of the stability of inorganic P pools in existing LSM is likely due to the calibration of models using net changes to inorganic soil P pools (Goll et al., 2017; Hou et al., 2019). As

the P exchange among two given pools is most often two-way, the calibration of residence times on the net exchange must lead to an underestimation of P turnover. The data provided here will allow to better calibrate soil P dynamics and/or evaluate modelled MRT.

Secondly, we found that residence times of P in slow inorganic pools varies considerably between soils, suggesting that land surface models should account for existing knowledge of P pool stability under different soil and environment conditions, rather than assuming globally uniform mean residence times. We found variation over several orders of magnitude in mean residence times of the same pool between different soils, and this variation could partly be explained by secondary soil and environment variables.

Thirdly, land surface models should consider that the HCl-P pool may have two-way exchange of P. Current model formulations assume the HCl-P pool contains only primary apatite, and the flux between the HCl-P pool and the soil solution is one-way (only dissolution) (Hou et al., 2019; Yang et al., 2014; Yu et al., 2018). From empirical observations it is known that HCl-P does not only contain primary apatite but may also contain secondary P minerals (Helfenstein et al., 2018a; Tamburini et al., 2012). Under acidic conditions, this may be Fe- and Al-associated minerals, while under neutral and alkaline pH this may be precipitated Ca-phosphates (Frossard et al., 1995). Also, P radioisotope transfer from the soil solution to the HCl-P pool is well-documented (Buehler et al., 2002; Pistocchi et al., 2018). To take this into account, future P model formulation should consider a two-way flux between the HCl-P pool and the soil solution, with an exchange rate dependent on pH (see Eq. 4). While this change is likely to have limited impact for the modelling of neutral to alkaline soils, where the HCl-P pool is stable, in more acidic soils the HCl-P pool tends to have similar dynamics to the NaOH-P pool, thus having considerable impact on P cycling and P availability.

In conclusion, mean residence times for inorganic P pools proposed here and the lessons learned should help improve model formulation of P in land surface models. Our estimates of MRTs suggest that current land surface models overestimate P MRT in inorganic soil P pools and as a consequence might overestimate the importance of biological soil P transformation (e.g. via phosphatases). Also, the temporal dynamics of P pools was found to vary largely between different soil types, which is not captured by models. Finally, model formulations should refrain from equating HCl-P to primary P, as this pool often contains secondary P minerals and is relatively dynamic in low pH soils. More empirical data on soil P pool mean residence times is needed, also from soil-plant systems and field experiments, but our data provides the basis to start building data-constrained soil P models.

**Data availability**

All data is available in the supplementary csv file "all_data.csv" except for isotope exchange kinetic data from the DOK experiment (5 soils). This data has not yet been published and is in the process of being published by Astrid Oberson. For isotope exchange kinetic data from the DOK experiment please contact Astrid Oberson, ETH Zürich, Switzerland.

**Author contributions**

Data collection, analysis and preparation of the manuscript were done by JH and CP in equal terms, with contributions from EF, AO, DG and FT.

**Competing interests**

The authors declare that they have no conflict of interest.

**Acknowledgements**

The authors would like to thank all primary data producers whose data was re-used in this study. We are also very grateful to Enqing Hou, Lin You,
Sebastian Doetterl, Peter Vitousek and an anonymous reviewer for providing valuable comments to improve the manuscript. Also, the authors are grateful to fruitful discussions with Janine Burren, Philippe Deleporte, Timothy McLaren, and Federico Cantini. Funding was provided by the Swiss National Science Foundation (project no. 200021_162422).

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

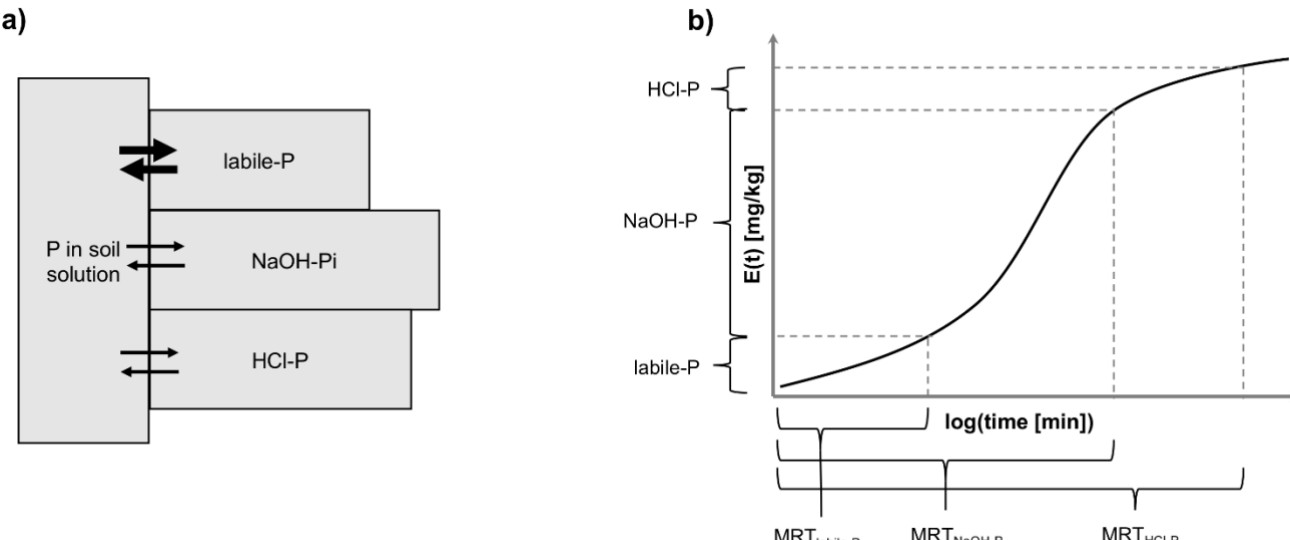

**Figure 1. Conceptual diagram showing: a) how Hedley P pools exchanges with the soil solution (modified after Fardeau et al. (1993)); b) how mean residence time was calculated. E(t) shows the amount of phosphate that has passed through the soil solution as a function of time.**

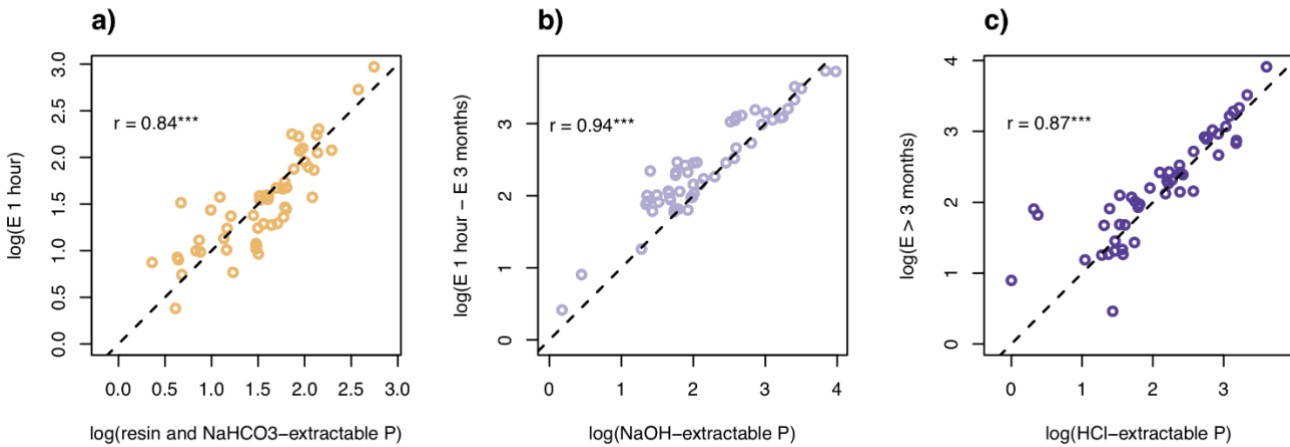

**Figure 2. Correlation between chemically extracted pools and isotopically exchangeable P. Resin and NaHCO₃-extractable P correlated with P exchangeable in 1 hour (E 1 hour) (a), NaOH-extractable P with P exchangeable between 1 hour and 3 months (b), and HCl-extractable P with P only exchangeable in over three months (c). Units of both axes are log(mg P kg-1). Dotted line shows the 1:1 line. Pearson's product-moment correlation (r value on plot) was highly significant (p < 0.001) for all three correlations.**

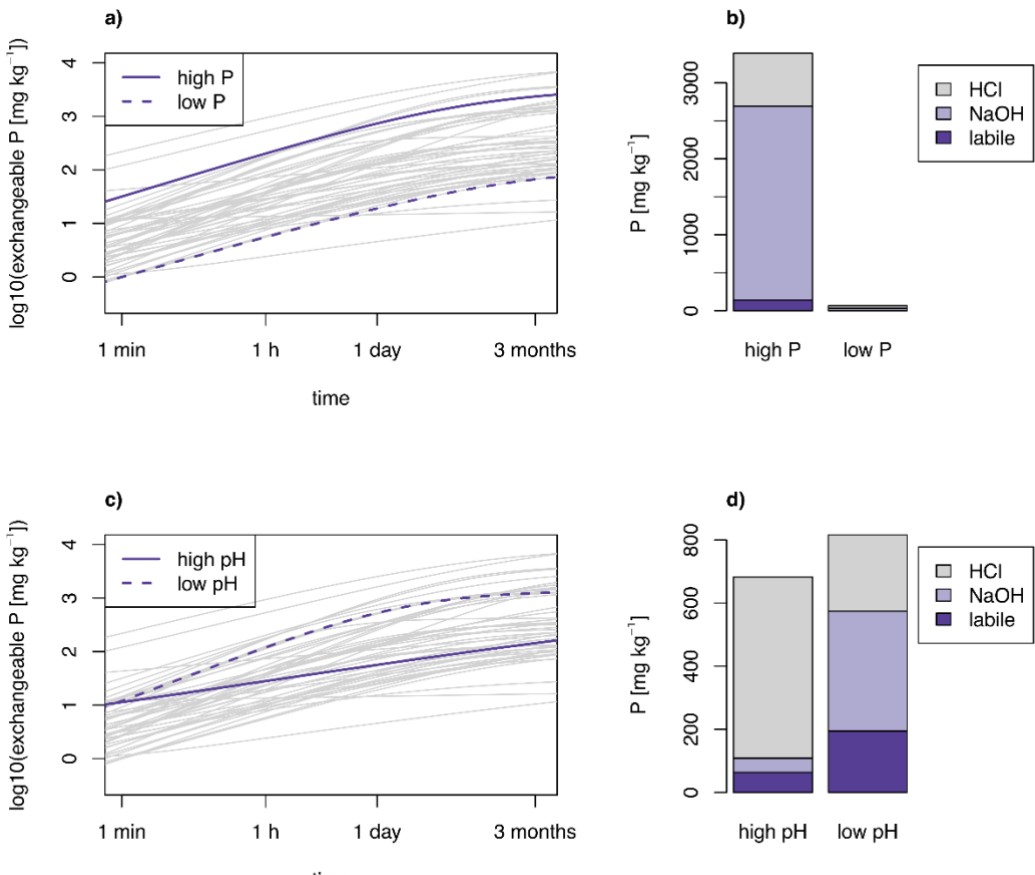

**Figure 3. Examples of exchangeable P as a function of time (E-curves). Grey lines show E-curves of each soil in the dataset. Panels (a) and (b) highlight two highly-reactive soils, one with high amounts of P (an Andosol from Helfenstein et al., 2018a) and one with little P (a Ferralsol from Oberson et al., 1999). (a) Shows the E-curves and (b) the corresponding sequential extraction (b). Panels (c) and (d) highlight two soils with similar amounts of total P, but different pH and P exchange behavior. For the high pH Fluvisol (pH = 8.1, from Borda et al., 2014), P-exchange is slow, compared to a Cambisol with much more exchange on the fast-intermediate term (pH = 3.8, from Lang et al., 2016) (c). In the high pH soil most P is HCl-extractable, whereas for the low pH soil more P is found in the NaOH and labile pools (d).**

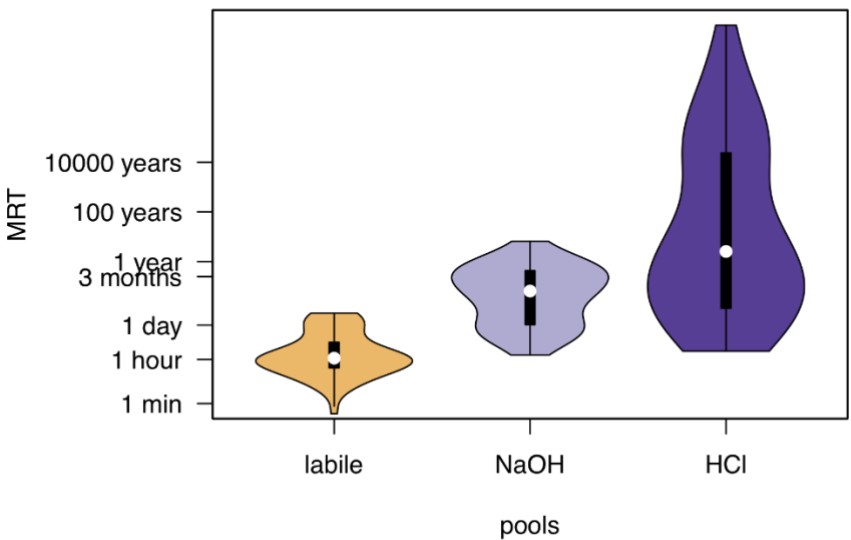

**Figure 4. Distribution of mean residence times of P in labile, NaOH and HCl pools for 53 soils. The black bars show a boxplot and the colored area shows the kernel density distribution.**

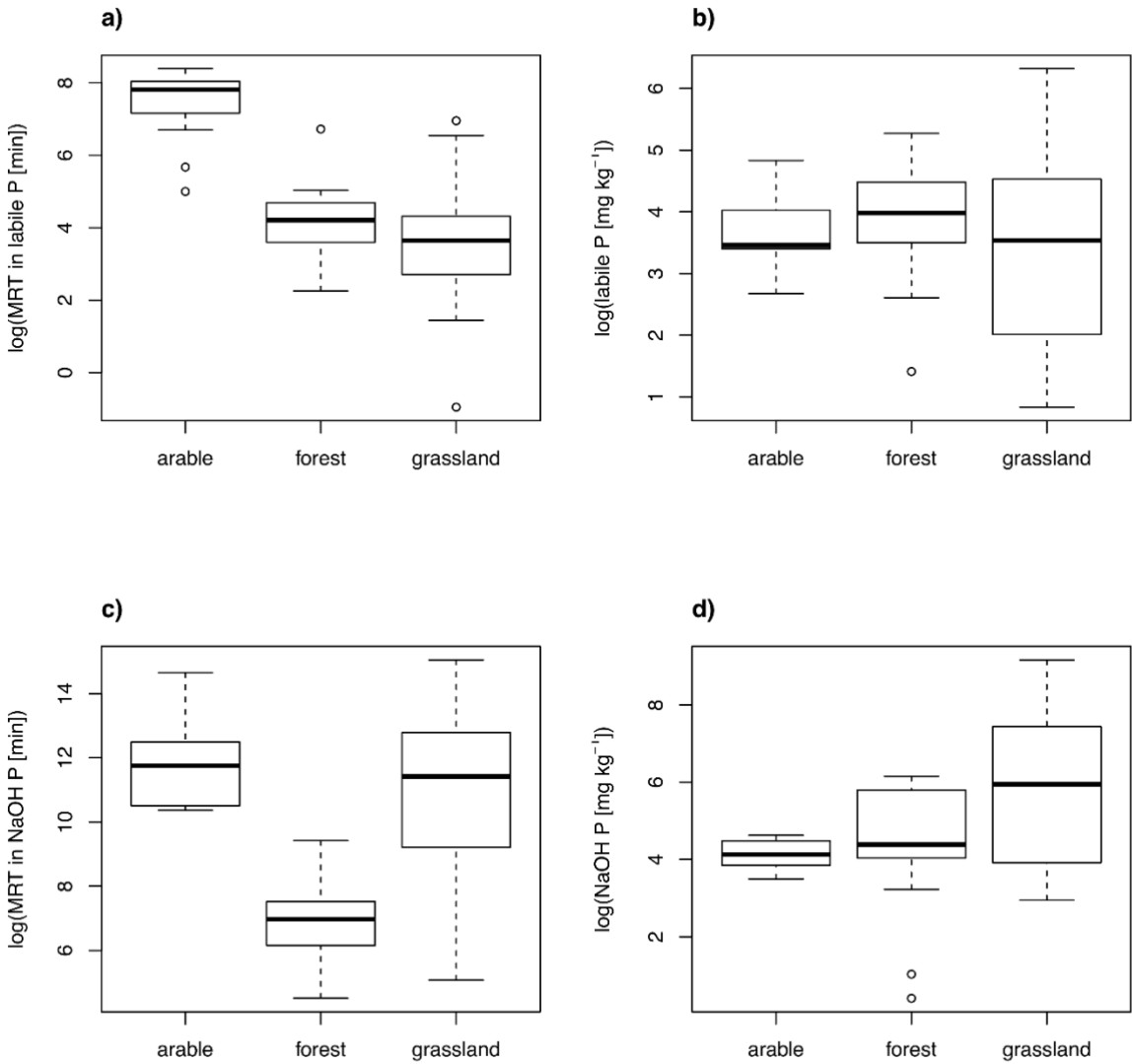

**Figure 5. Mean residence time (MRT) of P (a, c) and pool size (b, d) as a function of land use. Both MRT of P in labile P (a) and NaOH-P (c) was significantly affected by land use.**

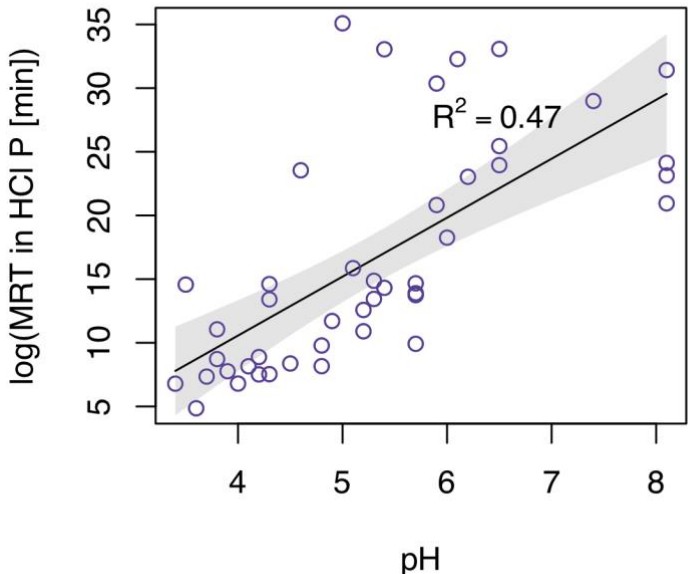

**Figure 6. Simple regression of calculated mean residence time of P in HCl-P with pH. *F*-statistic = 37.7, *p* < 0.001.**

**Table 1. Selected soil properties, climate conditions, soil phosphorus pools, and phosphorus exchange kinetic properties of soils used in this study.**

| | Variable | Description | min | quartile 1 | median | mean | quartile 3 | max | sample size |
|---|---|---|---|---|---|---|---|---|---|
| soil properties | pH | pH measured in water | 3.36 | 4.30 | 5.20 | 5.36 | 6.20 | 8.10 | 53 |
| | sand | sand content [g kg-1] | 80 | 190 | 360 | 359 | 430 | 800 | 53 |
| | silt | silt content [g kg-1] | 160 | 340 | 400 | 413 | 440 | 710 | 53 |
| | clay | clay content [g kg-1] | 40 | 160 | 220 | 228 | 290 | 399 | 53 |
| | organic C | organic C [g kg-1] | 11 | 23.5 | 46.5 | 60 | 80.2 | 230.4 | 53 |
| climate | MAT | mean annual. temperature [° C] | 4.5 | 8.4 | 13.0 | 14.0 | 19.1 | 27.0 | 53 |
| | MAP | mean annual. precipitation [mm] | 275 | 792 | 1046 | 1285 | 1578 | 3123 | 53 |
| soil P pools | total P | total P [mg kg-1] | 167 | 693 | 1016 | 2772 | 3764 | 20990 | 44 |
| | resin P | resin-extractable P [mg kg-1] | 0.90 | 6.95 | 22.98 | 42.55 | 45.84 | 385.93 | 42 |
| | NaHCO$_3$ P | NaHCO$_3$ -extractable P [mg kg-1] | 1.12 | 9.70 | 21.40 | 32.40 | 37.47 | 170.72 | 53 |
| | NaOH P | NaOH -extractable P [mg kg-1] | 1.5 | 48.0 | 99.4 | 744.6 | 470.4 | 9547.8 | 53 |
| | HCl P | HCl -extractable P [mg kg-1] | 1.0 | 34.5 | 162.5 | 433.1 | 557.5 | 4040.7 | 50 |

| | | | | | | | | | |
|---|---|---|---|---|---|---|---|---|---|
| | Pw | water-extractable P [mg kg$^{-1}$] | 0.013 | 0.328 | 1.00 | 3.782 | 2.6 | 42.5 | 53 |
| P exchange kinetics | m | exchange parameter | 0.01 | 0.06 | 0.15 | 0.26 | 0.38 | 0.97 | 53 |
| | n | exchange parameter | 0.03 | 0.40 | 0.46 | 0.45 | 0.50 | 0.76 | 53 |
| | E1min | P exchangeable in 1 minute [mg kg$^{-1}$] | 1.0 | 2.7 | 4.9 | 13.8 | 11.9 | 218.2 | 53 |
| | E3months | P exchangeable in 3 months [mg kg-1] | 12 | 111 | 251 | 806 | 1235 | 6311 | 53 |

**Table 2. Multiple regression models for mean residence times of P in labile P, NaOH-extractable P, and HCl-P. Models were determined by a step-wise selection process that maximizes Akaike's Information Criterion.**

| response variable | multiple regression model | adjusted R$_2$ | *F*-statistic | significance level |
|---|---|---|---|---|
| t_labile | log(t_labile) = 8.92 - 0.07 (clay) + (land use)[a] | 0.63 | 30.6 | < 0.001 |
| t_NaOH | log(t_NaOH) = 11.6 - 0.09 (clay) + 0.80 log(Corg) + (land use)[b] | 0.57 | 18.4 | < 0.001 |
| t_HCl | log(t_HCl) = -12.9 + 5.23 (pH) - 0.21 (clay) + 5.2 x 10$_{-3}$ (mean.rainfall) | 0.54 | 17.8 | < 0.001 |

[a]for arable = 0; for forest = -3.08; for grassland = -3.78

[b]for arable = 0; for forest = -5.94; for grassland = -1.81

Table 3. Mean residence times of different soils calculated in this study compared to values used in land surface models. Mean residence times of the resin and labile (resin + bicarbonate) P pools are displayed in minutes, while mean residence times of the NaOH and HCl P pools are displayed in years. Residence times of the listed models were extracted from the literature or by contacting respective authors (Wang et al. 2010; Goll et al. 2012; Yang et al. 2014; Zhu et al. 2016; Goll et al. 2017).

| soil order | resin P [min] | | | | labile P[a] [min] | | | | NaOH-P [yr] | | | | HCl-P [yr] | | | |
|---|---|---|---|---|---|---|---|---|---|---|---|---|---|---|---|---|
| | mean | sd | median | n | mean | sd | median | n | mean | sd | median | n | mean | sd | median | n |
| Acrisols | - | - | - | - | 93 | - | 93 | 1 | 0.47 | - | 0.47 | 1 | 1.8 | - | 1.8 | 1 |
| Andosols | 16.0 | 19.8 | 7.1 | 12 | 137 | 260 | 62 | 17 | 0.84 | 1.5 | 0.45 | 17 | $2.8 \times 10^8$ | $9.6 \times 10^8$ | $4.0 \times 10^4$ | 12 |
| Cambisols | 33.9 | 42.6 | 22.4 | 14 | 132 | 226 | 57 | 19 | 0.03 | 0.05 | 0.003 | 19 | 14 | 42.5 | 0.01 | 20 |
| Ferralsols | 6.1 | 4.5 | 5.2 | 7 | 77 | 106 | 27 | 7 | 0.05 | 0.09 | 0.002 | 7 | 1.4 | 1.9 | 0.75 | 4 |
| Fluvisols | 116.6 | 101.8 | 85.8 | 4 | $3.1 \times 10^3$ | $1 \times 10^3$ | $3.0 \times 10^3$ | 4 | 0.12 | 0.11 | 0.07 | 4 | $2.1 \times 10^7$ | $4.2 \times 10^7$ | $3.9 \times 10^4$ | 4 |
| Luvisols | 78.7 | 28.2 | 63.5 | 5 | $2.5 \times 10^3$ | $1 \times 10^3$ | $2.6 \times 10^3$ | 5 | 1.3 | 1.8 | 0.60 | 5 | $1.6 \times 10^5$ | $2.0 \times 10^5$ | $1.1 \times 10^5$ | 4 |
| all soils | 37.4 | 51.4 | 19.0 | 42 | 571 | $1.1 \times 10^3$ | 67 | 53 | 0.42 | 1.08 | 0.07 | 53 | $7.8 \times 10^7$ | $5.0 \times 10^8$ | 2.6 | 44 |

| models | plant-available / fast pool [min] | intermediate pool [yr] | slow pool [yr] |
|---|---|---|---|
| CABLE (Wang et al. 2010) | variable, dynamic equilibrium | 150 | 150 |
| JSBACH (Goll et al. 2012) | variable, dynamic equilibrium | 150 | - |
| ELM-CTC (Yang et al. 2014) | variable, dynamic equilibrium | 21-83 | $8.3 \times 10^4$ |
| ELM-ECA (Zhu et al. 2016) | variable, dynamic equilibrium | - | $8.3 \times 10^4$ |
| ORCHIDEE-NP (Goll et al. 2017) | variable, dynamic equilibrium | 25 | - |

[a] resin and bicarbonte-extractabe P.