# Peer review of "Estimates of mean residence times of phosphorus in commonly-considered inorganic soil phosphorus pools"

_Biogeosciences, 2019_

## Referee Comment (RC1) · Anonymous Referee #2 · 30 Aug 2019

Helfenstein et al. propose a method to calculate the mean residence time (MRT) of inorganic phosphorus (P) in different soil inorganic P pools based on data from short-term isotope exchange data. The short-term isotope exchange data is gained in isotope exchange studies that have been conducted in order to estimate the capacity of the soils to sorb P. The experiments typically last for 90 minutes and during this time the 33P activity in the soil solution decreases strongly due to sorption to the soil solid phase. Isotope exchange experiments have been used by Frossard et al. and other working groups to learn about the plant availability of different P pools in soil. I have doubts about the validity of the approach proposed here for the following reasons. First, the calculation of MRT is based on the assumption of steady state (inputs=outputs).

[Figure]

However, it is highly questionable whether this assumption is correct. In the experiments used by the authors, typically 10 g of sieved soil are oversaturated by 100 ml of water and put on a shaker until the end of the experiment. Being oversaturated with water and kept on a shaker is not a natural condition for soils. Thus, very likely the soils move away from steady state during these experiments due to this disturbance. Thus, the assumption of study state is very likely violated in the experiments, on which the calculation is based. Second, the authors derive MRTs of more than 10,000 years (Fig. 4) from experiments that last only for a much, much shorter period of time, which is problematic since the exchange processes measured over a short period of time are extrapolated by a factor of > 1,000,000. Third, the calculation proposed by the authors (Equation 1) is based on the assumption that the specific 33P activity of the soils solution (33P in solution to total inorganic P in solution) is stable during the 90-minutes experiment. This assumption is not correct. During 90-minutes experiments the specific 33P activity of the decreases exponentially, which means that the ratio of 33P and total inorganic P decreases, which is mostly due to adsorption of 33P to the soil solid phase. This change of the specific 33P activity is problematic because 33P is used as a tracer, and the ratio of the tracer to the inorganic P in the soil solution is changing continuously over the course of a short-term experiment. The proposed approach would require that the ratio of 33P to inorganic P in the solution is stable, but the specific 33P activity of the soil solution approaches a stable value only over longer periods of time. In addition (and less importantly), the description of the applied calculation is incomplete, and the calculation of the variables n and m, which are derived from short-term experiments is not explained in the present manuscript.
* * *

---

## Author Comment (AC1) · 4 Sep 2019

We thank the reviewer for their comments and would like to address the questions raised by the reviewer point by point.

"The short-term isotope exchange data is gained in isotope exchange studies that have been conducted in order to estimate the capacity of the soils to sorb P. The experiments typically last for 90 minutes and during this time the 33P activity in the soil solution decreases strongly due to sorption to the soil solid phase. Isotope exchange experiments have been used by Frossard et al. and other working groups to learn about the plant availability of different P pools in soil."

[Figure]

Isotope exchange kinetic experiments (IEK) were developed to "determine the rate at which phosphate ions in the soil solution are renewed from phosphate ions in the solid phase," not to measure soil P sorption capacity (Fardeau et al. 1991). Isotopic methods using P radioisotopes date back to the 1940s (McAuliffe et al. 1948). IEK method was then further developed and tested by J.C. Fardeau and colleagues (Fardeau and Marini 1968, Fardeau and Jappe 1988, Fardeau 1985). Unfortunately, these and further key publications validating the IEK method are in French, which explains why they are little known outside of French-speaking regions. IEK experiments are thus based on, and have done much to further, our understanding of exchange mechanisms. This explains why E-values derived from IEK perform much better at predicting crop response than other soil P tests (Frossard et al. 1994), and why E-values are widely accepted as the gold standard for determining P bioavailability (Hamon et al. 2002, Kruse et al. 2015).

"I have doubts about the validity of the approach proposed here for the following reasons. First, the calculation of MRT is based on the assumption of steady state (inputs=outputs). However, it is highly questionable whether this assumption is correct. In the experiments used by the authors, typically 10 g of sieved soil are oversaturated by 100 ml of water and put on a shaker until the end of the experiment. Being oversaturated with water and kept on a shaker is not a natural condition for soils. Thus, very likely the soils move away from steady state during these experiments due to this disturbance. Thus, the assumption of study state is very likely violated in the experiments, on which the calculation is based."

It is true, as the reviewer pointed out, that the soil/water suspension we use in IEK is not similar to soil natural conditions, however several studies comparing E-values extrapolated from IEK and L-values from pot experiments with plants (so under conditions much closer to "natural") have shown that these two independent estimations of P exchangeability closely overlap. Fardeau and Jappe (1976), showed that measured L-values and extrapolated E-values were very similar. Morel and Plenchette (1994) showed the same for soybean and barley (measuring E-values on soil suspensions in

periods from 1 min to several days) and Frossard et al. (1994) for Agrostis on several different soil types. Similarly, Sinaj et al., (2004) demonstrated the same for zinc (different nutrient, but same IEK method) on several different soils and plant species. The comparisons between E-values extrapolated from IEK experiments and L-values measured in pot experiments with plants suggest that extrapolated E-values are a good approximation for P exchange between solid phase and soil solution. Although in IEKs soil is suspended in water, the soil/solution transfer is not readily affected, especially compared to other methods using chemical extractions.

"Second, the authors derive MRTs of more than 10,000 years (Fig. 4) from experiments that last only for a much, much shorter period of time, which is problematic since the exchange processes measured over a short period of time are extrapolated by a factor of > 1,000,000."

We agree that estimates of longer MRTs (i.e. for the HCl-P pool) are more uncertain because extrapolated E-values can only be validated with incubation data for time spans up to several months due to the short half-life of the P radioisotopes. This limitation is clearly stated in the manuscript (p. 9, l. 19). However, three independent lines of evidence support our estimates even of the very long MRTs of HCl-P. Firstly, the strong correlation (following the 1:1 line) of E-values with sequentially extracted pools, also for P not exchangeable in three months and HCl-P (Fig. 2). This shows that these pools are clearly related even though measured by independent methods. The same correlation was found already by Frossard et al (1996) for a sewage sludge in which the HCl-P pool was composed by sparsely soluble (and therefore slowly exchangeable) Ca-P minerals. Secondly, stable oxygen isotopic ratios of phosphate have been studied in HCl-P pools in soils of known ages, and shown that indeed this pool may remain stable for time spans of years to » millennia, depending on environmental conditions and soil properties (p. 9, l. 23-32). This range fits with our estimates. See (p. 9, l. 23-32). Thirdly, the strong relationship between pH and MRT of HCl-P that we observed in our data is supported by empirical observation, or at least make sense given what

we know about the composition and stability of HCl-P (p. 8, l. 19-24). Hence, both the order of magnitude and the pattern in our HCl-P estimates make sense given what we know about P species extracted by HCl and their relative stability under different environmental conditions.

"Third, the calculation proposed by the authors (Equation 1) is based on the assumption that the speciïñÄc 33P activity of the soils solution (33P in solution to total inorganic P in solution) is stable during the 90-minutes experiment. This assumption is not correct. During 90-minutes experiments the speciïñÄc 33P activity of the decreases exponentially, which means that the ratio of 33P and total inorganic P decreases, which is mostly due to adsorption of 33P to the soil solid phase. This change of the speciïñÄc 33P activity is problematic because 33P is used as a tracer, and the ratio of the tracer to the inorganic P in the soil solution is changing continuously over the course of a short-term experiment. The proposed approach would require that the ratio of 33P to inorganic P in the solution is stable, but the speciïñÄc 33P activity of the soil solution approaches a stable value only over longer periods of time."

We do not understand the reviewer's concern here: equation 1 is not based on the assumption that specific activity of 33P (or 32P) is constant in the soil solution. Rather, the denominator of Eq. 1 specifically describes the change in specific activity. IEK experiments provide insights on P exchange between the soil solution and the solid phase by measuring the decrease of radioisotope activity in the soil solution. This decrease in specific activity is modelled by $r(t)/R = m(t + m^{(1/n)})^{(-n)} + r_{(\infty)}/R$ Where $r(t)$ is the radioactivity (Bq) measured at time $t$ (min), R is the total amount of radioactivity added, and m and n are the model parameters that describe the rapid and slow physicochemical processes, respectively (Fardeau et al. 1991, Frossard et al. 2011).

"In addition (and less importantly), the description of the applied calculation is incomplete, and the calculation of the variables n and m, which are derived from short-term experiments is not explained in the present manuscript."

We thank the reviewer for pointing out that the description needs to be improved. In the present version we kept the description of Eq. 1 and the meaning of the variables m and n minimal because it has been described in detail in previous publications (Fardeau et al. 1991, Frossard et al. 2011, Helfenstein et al. 2018, ...). To address the reviewers concern, we propose to expand the background information on Eq. 1 in the revision.

References Fardeau JC, Jappe J (1988) Valeurs caractéristique des cinétiques de dilution isotopique des ions phosphate dans les systèmes sol-solution. In: Gachon L (ed) Phosphore et potassium dans les relations sol-plante. Institute National de la Recherche Agronomique, Paris, pp 79–99

Fardeau J-C, Jappe J (1976) Nouvelle methode de determination du phosphore du sol assimilable par les plantes: extrapolation des cinetiques de dilution isotopique. C R, Ser D 282:1137–1140

Fardeau JC, Marini P (1968) Sur la détermination des cinétiques d'échange isotopique des ions-phosphate d'un sol. CR Acad Sci Paris

Fardeau JC (1985) Cinétique d'échange des ions phosphate dans les systèmes sol-solution. Vérification expérimentale de l'équation théorique. CR Acad Sci Paris t 300:371–376

Fardeau J-C, Morel C, Boniface R (1991) Phosphate ion transfer from soil to soil solution: kinetic parameters. Agronomie 11:787–797

Frossard, E., Morel, J.L., Fardeau, J.C., Brossard, M., 1994. Soil Isotopically Exchangeable Phosphorus: A Comparison between E and L Values. Soil Science Society of America Journal 58, 846–851.

Frossard, Emmanuel, Achat, D.L., Bernasconi, S.M., Bunemann, E.K., Fardeau, J.C., Jansa, J., Morel, C., Rabeharisoa, L., Randriamanantsoa, L., Sinaj, S., Tamburini, F., Frossard, E., Oberson, A., 2011. The Use of Tracers to Investigate Phosphate Cycling in Soil–Plant Systems, in: Phosphorus in Action, Soil Biology. Bunemann E. K., Berlin

Heidelberg, pp. 59–91.

Frossard, E., Sinaj, S., Dufour, P., 1996. Phosphorus in urban sewage sludges as assessed by isotopic exchange. Soil Science Society of America Journal 60, 179–182.

Hamon RE, Bertrand I, McLaughlin MJ (2002) Use and abuse of isotopic exchange data in soil chemistry. Soil Res 40:1371–1381.

Helfenstein J, Jegminat J, McLaren TI, Frossard E (2018) Soil solution phosphorus turnover: derivation, interpretation, and insights from a global compilation of isotope exchange kinetic studies. Biogeosciences 15:105–114.

Kruse J, Abraham M, Amelung W, et al (2015) Innovative methods in soil phosphorus research: A review. J Plant Nutr Soil Sci 178:43–88.

Morel C, Plenchette C (1994) Is the isotopically exchangeable phosphate of a loamy soil the plant-available P? Plant Soil 158:287–297.

Sinaj, S., Dubois, A., Frossard, E., 2004. Soil isotopically exchangeable zinc: A comparison between E and L values. Plant and Soil 261, 17–28.

---

## Referee Comment (RC2) · Enqing Hou (Referee) · 30 Sep 2019

The estimate of MRT deserves an encouragement, because it provides a fundamental and quantitative insight into the dynamics of P in soil. In this regard, I would like to support the publication of this work. However, before this, I have some concerns for the authors to address. I am curious about why the MRT of HCl-P can be estimated in the way used by the authors. HCl-P is mainly of apatite P in neutral and alkaline soils. The dynamics of apatite P should be unidirectional, that is apatite P is always depleted without a formation during the experimental duration (90 mins). So, an exchangeable between resin P and HCl P is unreasonable at least for neutral and alkaline soils (Fig.

1), although it's possible if HCl P is largely of Fe/Al associated P, as in acidic soils. All the estimates of MRT are obtained based on laboratory incubation. The estimates therefore should be much different from those in field, which can be affected by soil moisture and temperature and many other factors. This limitation and their potential effects on the estimates should be noted. Are the estimates comparable to the estimates of Hou et al. (2019) based on greenhouse experiments? The Figure 1 and the calculation of MRT of NaOH-P and HCl-P are weird. MRT-NaOHP is estimated based on the sum of labile P and NaOH-P (while not only NaOH alone?). MRT-HCIP is estimated on the sum of labile P, NaOH-P, and HCl P (while not on HCl P alone?). This will at least confuse readers, which do MRT-NaOHP and MRT-HCl really measure? In Fig. 1. the conceptual diagram differs from many other diagrams, such as Hou et al. JGR Biogeosciences (2019) and Tiessen et al. (1984). The model structure (conceptual diagram) affects the estimate of MRT. I think this should be discussed to let readers know there are other possible exchange pathways among soil P pools that will affect the estimate of MRTs. Give units in Figs. 2 and 5. Give Y axis lab (MRT?) in Fig. 4 In L15, "and call these soil P pools", I think I understand what you mean, but it reads a bit weird. L24-25: which two studies? Does the filled data affect much of the results? I think the data used by the authors are valuable. Why not make the raw data and the fitted m and n values open access?

---

## Author Comment (AC2) · 6 Nov 2019

*We thank the reviewer for their comments and would like to address the questions raised by the reviewer point by point. In this version of the response to the reviewer, we also outline where we made changes in the manuscript in response to the reviewer comments.*

"The short-term isotope exchange data is gained in isotope exchange studies that have been conducted in order to estimate the capacity of the soils to sorb P. The experiments typically last for 90 minutes and during this time the 33P activity in the soil solution decreases strongly due to sorption to the soil solid phase. Isotope exchange experiments have been used by Frossard et al. and other working groups to learn about the plant availability of different P pools in soil."

*Isotope exchange kinetic experiments (IEK) were developed to "determine the rate at which phosphate ions in the soil solution are renewed from phosphate ions in the solid phase," not to measure soil P sorption capacity (Fardeau et al. 1991). Isotopic methods using P radioisotopes date back to the 1940s (McAuliffe et al. 1948). IEK method was then further developed and tested by J.C. Fardeau and colleagues (Fardeau and Marini 1968, Fardeau and Jappe 1988, Fardeau 1985). Unfortunately, these and further key publications validating the IEK method are in French, which explains why they are little known outside of French-speaking regions.*

*IEK experiments are thus based on, and have done much to further, our understanding of exchange mechanisms. This explains why E-values derived from IEK perform much better at predicting crop response than other soil P tests (Frossard et al. 1994), and why E-values are widely accepted as the gold standard for determining P bioavailability (Hamon et al. 2002, Kruse et al. 2015).*

"I have doubts about the validity of the approach proposed here for the following reasons. First, the calculation of MRT is based on the assumption of steady state (inputs=outputs). However, it is highly questionable whether this assumption is correct. In the experiments used by the authors, typically 10 g of sieved soil are oversaturated by 100 ml of water and put on a shaker until the end of the experiment. Being oversaturated with water and kept on a shaker is not a natural condition for soils. Thus, very likely the soils move away from steady state during these experiments due to this disturbance. Thus, the assumption of study state is very likely violated in the experiments, on which the calculation is based."

*It is true, as the reviewer pointed out, that the soil/water suspension we use in IEK is not similar to soil natural conditions, however several studies comparing E-values extrapolated from IEK and L-values from pot experiments with plants (so under conditions much closer to "natural") have shown that these two independent estimations of P exchangeability closely overlap. Fardeau and Jappe (1976), showed that measured L-values and extrapolated E-values were very similar. Morel and Plenchette (1994) showed the same for soybean and barley (measuring E-values on soil suspensions in periods from 1 min to several days) and Frossard et al. (1994) for Agrostis on several different soil types. Similarly, Sinaj et al., (2004) demonstrated the same for zinc (different nutrient, but same IEK method) on several different soils and plant species.*

*The comparisons between E-values extrapolated from IEK experiments and L-values measured in pot experiments with plants suggest that extrapolated E-values are a good approximation for P exchange between solid phase and soil solution. Although in IEKs soil is suspended in water, the soil/solution transfer is not readily affected, especially compared to other methods using chemical extractions.*

*Finally, in the discussion section (p. 10 l. 26 to p. 11 l. 9) we validate our estimates against incubation experiments and field measurements.*

*To state the limitation more clearly of measuring MRTs in simplified laboratory conditions, we added the following sentences to the "limitations" section (p. 12, l. 2-6) : "Our approach only considers a simplified system of soil and water in steady-state conditions, and excludes biological activity. In field conditions, P residence times may be different due to non-steady state conditions, microbial interactions with abiotic processes, as well as plant uptake and alterations of the physical and chemical soil environment (Hinsinger, 2001). For example, intensive P uptake by plants may lead to net changes in soil P pools in addition to exchange fluxes (Guo et al., 2000). Also, it is likely that MRTs are affected by temperature and changes in soil moisture."*

"Second, the authors derive MRTs of more than 10,000 years (Fig. 4) from experiments that last only for a much, much shorter period of time, which is problematic since the exchange processes measured over a short period of time are extrapolated by a factor of > 1,000,000."

*We agree that estimates of longer MRTs (i.e. for the HCl-P pool) are more uncertain because extrapolated E-values can only be validated with incubation data for time spans up to several months due to the short half-life of the P radioisotopes. This limitation is clearly stated in the manuscript (p. 1, l. 6-9). However, three independent lines of evidence support our estimates even of the very long MRTs of HCl-P.*

*Firstly, the strong correlation (following the 1:1 line) of E-values with sequentially extracted pools, also for P not exchangeable in three months and HCl-P (Fig. 2). This shows that these pools are clearly related even though measured by independent methods. The same correlation was found already by Frossard et al (1996) for a sewage sludge in which the HCl-P pool was composed by sparsely soluble (and therefore slowly exchangeable) Ca-P minerals.*

*Secondly, stable oxygen isotopic ratios of phosphate have been studied in HCl-P pools in soils of known ages, and shown that indeed this pool may remain stable for time spans of years to >> millennia, depending on environmental conditions and soil properties. This range fits with our estimates. See (p. 11, l. 11-19).*

*Thirdly, the strong relationship between pH and MRT of HCl-P that we observed in our data is supported by empirical observation, or at least make sense given what we know about the composition and stability of HCl-P (p. 10, l. 1-5). Hence, both the order of magnitude and the pattern in our HCl-P estimates make sense given what we know about P species extracted by HCl and their relative stability under different environmental conditions.*

"Third, the calculation proposed by the authors (Equation 1) is based on the assumption that the specific 33P activity of the soils solution (33P in solution to total inorganic P in solution) is stable during the 90-minutes experiment. This assumption is not correct. During 90-minutes experiments the specific 33P activity of the decreases exponentially, which means that the ratio of 33P and total inorganic P decreases, which is mostly due to adsorption of 33P to the soil solid phase. This change of the specific 33P activity is problematic because 33P is used as a tracer, and the ratio of the tracer to the inorganic P in the soil solution is changing continuously over the

course of a short-term experiment. The proposed approach would require that the ratio of 33P to inorganic P in the solution is stable, but the specific 33P activity of the soil solution approaches a stable value only over longer periods of time."

*We do not understand the reviewer's concern here: equation 1 is not based on the assumption that specific activity of 33P (or 32P) is constant in the soil solution. Rather, the denominator of Eq. 1 specifically describes the change in specific activity. IEK experiments provide insights on P exchange between the soil solution and the solid phase by measuring the decrease of radioisotope activity in the soil solution. This decrease in specific activity is modelled by*

$$\frac{r(t)}{R} = m\left(t + m^{\frac{1}{n}}\right)^{-n} + \frac{r_{(\infty)}}{R}$$

*Where r(t) is the radioactivity (Bq) measured at time t (min), R is the total amount of radioactivity added, and m and n are the model parameters that describe the rapid and slow physicochemical processes, respectively (Fardeau et al. 1991, Frossard et al. 2011).*

"In addition (and less importantly), the description of the applied calculation is incomplete, and the calculation of the variables n and m, which are derived from short-term experiments is not explained in the present manuscript."

*We thank the reviewer for pointing out that the description needs to be improved. We added the following sentence "The parameters m and n describe the rapid and slow physicochemical exchange processes, respectively, and are determined by fitting a non-linear regression model to measurements of radioisotope concentration in solution (for details see Fardeau et al. 1991 and Frossard et al. 2011)." (p. 5, l. 6-9). Based on the second reviewers' comments, we also added a file containing all of the primary data (including n and m values).*

**References**

Fardeau JC, Jappe J (1988) Valeurs caractéristique des cinétiques de dilution isotopique des ions phosphate dans les systèmes sol-solution. In: Gachon L (ed) Phosphore et potassium dans les relations sol-plante. Institute National de la Recherche Agronomique, Paris, pp 79–99

Fardeau J-C, Jappe J (1976) Nouvelle methode de determination du phosphore du sol assimilable par les plantes: extrapolation des cinetiques de dilution isotopique. C R, Ser D 282:1137–1140

Fardeau JC, Marini P (1968) Sur la détermination des cinétiques d'échange isotopique des ions-phosphate d'un sol. CR Acad Sci Paris

Fardeau JC (1985) Cinétique d'échange des ions phosphate dans les systèmes sol-solution. Vérification expérimentale de l'équation théorique. CR Acad Sci Paris t 300:371–376

Fardeau J-C, Morel C, Boniface R (1991) Phosphate ion transfer from soil to soil solution: kinetic parameters. Agronomie 11:787–797

Frossard, E., Morel, J.L., Fardeau, J.C., Brossard, M., 1994. Soil Isotopically Exchangeable Phosphorus: A Comparison between E and L Values. Soil Science Society of America Journal 58, 846–851. doi:10.2136/sssaj1994.03615995005800030031x

Frossard, Emmanuel, Achat, D.L., Bernasconi, S.M., Bunemann, E.K., Fardeau, J.C., Jansa, J., Morel, C., Rabeharisoa, L., Randriamanantsoa, L., Sinaj, S., Tamburini, F., Frossard, E., Oberson, A., 2011. The Use of Tracers to Investigate Phosphate Cycling in Soil–Plant Systems, in: Phosphorus in Action, Soil Biology. Bunemann E. K., Berlin Heidelberg, pp. 59–91.

Frossard, E., Sinaj, S., Dufour, P., 1996. Phosphorus in urban sewage sludges as assessed by isotopic exchange. Soil Science Society of America Journal 60, 179–182.

Hamon RE, Bertrand I, McLaughlin MJ (2002) Use and abuse of isotopic exchange data in soil chemistry. Soil Res 40:1371–1381.

Helfenstein J, Jegminat J, McLaren TI, Frossard E (2018) Soil solution phosphorus turnover: derivation, interpretation, and insights from a global compilation of isotope exchange kinetic studies. Biogeosciences 15:105–114.

Kruse J, Abraham M, Amelung W, et al (2015) Innovative methods in soil phosphorus research: A review. J Plant Nutr Soil Sci 178:43–88.

Morel C, Plenchette C (1994) Is the isotopically exchangeable phosphate of a loamy soil the plant-available P? Plant Soil 158:287–297.

Sinaj, S., Dubois, A., Frossard, E., 2004. Soil isotopically exchangeable zinc: A comparison between E and L values. Plant and Soil 261, 17–28.

---

## Author Comment (AC3) · 6 Nov 2019

"The estimate of MRT deserves an encouragement, because it provides a fundamental and quantitative insight into the dynamics of P in soil. In this regard, I would like to support the publication of this work. However, before this, I have some concerns for the authors to address."

*Thank you very much for taking the time to review our manuscript and for your supportive and constructive comments. Below, we have addressed your comments point by point and described our changes to the revised manuscript.*

"I am curious about why the MRT of HCl-P can be estimated in the way used by the authors. HCl-P is mainly of apatite P in neutral and alkaline soils. The dynamics of apatite P should be unidirectional, that is apatite P is always depleted without a formation during the experimental duration (90 mins). So, an exchangeable between resin P and HCl P is unreasonable at least for neutral and alkaline soils (Fig. 1), although it's possible if HCl P is largely of Fe/Al associated P, as in acidic soils."

*Indeed, exchange between HCl- P and the soil solution is often considered to be unidirectional in model formulations, i.e. a slow one-way flow of phosphates from the HCl-P pool to the soil solution via dissolution. This holds effectively true if we consider the net change of this pool over a pedological time scale (e.g. the Walker and Syers model). However, as pointed out by the reviewer, HCl-P may also contain secondary P forms, such as Fe/Al associated P. Even in neutral/alkaline soils, Ca-P forms extracted by HCl may be secondary because phosphate ions easily precipitate with Ca in systems containing carbonates (Frossard et al., 1995).*

*Analysis of stable oxygen isotopic ratios in phosphate have also confirmed the dynamics of the HCl-P pool. There, on all soils studied except for young/unweathered soils, oxygen in phosphate carried the biological signature, suggesting re-precipitation of phosphate from the soil solution into the HCl-P pool after cycling through the biosphere (Helfenstein et al., 2018; Tamburini et al., 2012). Similarly, radioisotopic tracing shows radioisotope tracer incorporation over time scales of days-weeks into HCl-P (Buehler et al., 2002). Due to the empirical evidence, we propose that future P model formulation should consider bi-directional exchange between HCl-P and the soil solution.*

*We explain why we propose bi-directional exchange in a new section in the discussion (p. 13, l. 10-18).*

"All the estimates of MRT are obtained based on laboratory incubation. The estimates therefore should be much different from those in field, which can be affected by soil moisture and temperature and many other factors. This limitation and their potential effects on the estimates should be noted. Are the estimates comparable to the estimates of Hou et al. (2019) based on greenhouse experiments?"

*We agree that MRTs in field situations are likely to be vary from MRT estimates based on laboratory experiments. Hou et al. 2019 calculated MRTs of P in P pools based on changes in P pool sizes over time with data from a pot experiment growing plants on 8 different soils (Guo et*

*al., 2000). Below we compare the MRT of Hou et al. 2019, calculated as the inverse of the reported turnover rates for the respective pools, to the median MRTs of our study (see Table 3).*

| Pool | MRT in Hou et al. 2019 (means) | MRT in our study (medians) |
|------|-------------------------------|----------------------------|
| Labile P | 25 days | << 1 day |
| NaOH-Pi | 43.5 days | 25 days |
| HCl-P | 3.1 years | 2.6 years |

*While MRT estimates for NaOH-Pi and HCl-P are similar, the range in MRTs was much larger in our study than in Hou et al. (2019). Also, the MRT for labile P is one order of magnitude lower in Hou et al. (2019) than in our study. These differences are not surprising given the different approaches used to estimate MRT of P in these pools. By measuring changes in P pool sizes, only net P fluxes are considered. The radioisotopic approach on the other hand also measures exchange fluxes not leading to changes in pool sizes. This is important because sorption/desorption fluxes without net changes in P pool sizes are the main process driving P cycling between the soil solution and inorganic P pools, and thus important for P bioavailability* (Frossard et al., 2000; Syers et al., 2008).

*To address the reviewer's concerns, we expanded on the discussion of limitations arising from laboratory estimates (p. 11, l. 3-5). Also, we point out how our estimates are different from MRT estimates considering net changes in P pool sizes (p. 3, l. 14-16).*

"The Figure 1 and the calculation of MRT of NaOH-P and HCl-P are weird. MRT-NaOHP is estimated based on the sum of labile P and NaOH-P (while not only NaOH alone?). MRT-HClP is estimated on the sum of labile P, NaOH-P, and HCl P (while not on HCl P alone?). This will at least confuse readers, which do MRT-NaOHP and MRT-HCl really measure?"

*We realize that the calculation was not properly described, making the understanding of the approach difficult. Hence, we added the following sentence: "The summation of more labile pools to estimate MRT of more recalcitrant pools is necessary because in this model "slow" exchanging pools can only exchange once "faster" exchanging pools have fully exchanged." (p. 6, l. 2-3). The explanation is followed by the justification of this assumption (p. 6, l. 6-9).*

"In Fig. 1 the conceptual diagram differs from many other diagrams, such as Hou et al. JGR Biogeosciences (2019) and Tiessen et al. (1984). The model structure (conceptual diagram) affects the estimate of MRT. I think this should be discussed to let readers know there are other possible exchange pathways among soil P pools that will affect the estimate of MRTs."

*We realize that one of the difficulties in P modelling is that many empirical studies and also many models are based on slightly different conceptual diagrams of P cycling. In preparing this paper, we had a lot of discussions with Daniel Goll (as a representative of the P modelling community and co-author on this paper) on how our approach relates to existing model formulations. Our approach has been to clearly state the assumptions (p. 6, l. 6-22) and discuss the limitations (p. 11, l. 22- p. 12, l. 9). Also, we made sure that our conclusions go beyond our conceptual model and apply to all P model formulations.*

*To address the reviewer's comments, we added an additional sentence to make it explicit that we did not consider exchange between NaOH-Pi and "occluded pools" (p. 6, l. 10-12).*

*Also, we adapted the concluding paragraphs to make them more appropriate also to other conceptual model formulations of P cycling (see section "implications for land surface modelling").*

"Give units in Figs. 2 and 5. Give Y axis lab (MRT?) in Fig. 4"

*Thank you for spotting this. We added units to the x- and y-axes in Fig 2 (in the figure legend because the axes labels are already crowded). Figure 5 already has units (no changes made). In Fig. 4 the y-axis is indeed MRT. We added that.*

"In L15, "and call these soil P pools", I think I understand what you mean, but it reads a bit weird."

*We deleted the sentence fragment because it is unnecessary and can lead to confusion.*

"L24-25: which two studies? Does the filled data affect much of the results?"

*The two studies that didn't have information on soil texture were Borda et al. 2014 and Helfenstein et al. 2018. This is listed in supplementary table 1, "Sources for data on soil and other properties for each site". Additionally, we now referenced the studies in the sentence (p. 4, l. 11).*

*To test if the "filled data" affects the results, we repeated the analysis while excluding the Borda and Helfenstein samples (Fig 1 below. Compare to supplementary figure 3). The F and p values changed slightly, but the general conclusion is the same: significant regressions for MRT of labile P and NaOH-P, but not for HCl-P, though all models point at the same trend. Because the "filled data" does not affect the results we made no changes in the manuscript.*

[Figure]

*Figure 1. Simple regression of calculated mean residence time of P with clay concentration, excluding samples for which soil texture data was taken from SoilGrids. The model values changed slightly, but show the same trend. For labile P (F-statistic = 5.4, p = 0.03) and NaOH-P (F-statistic = 9.8, p < 0.01), but not for HCP (F-statistic = 3.8, p = 0.06).*

"I think the data used by the authors are valuable. Why not make the raw data and the fitted m and n values open access?"

*Thank you for finding our data valuable. We will provide it as a supplementary csv table in the revision. See "all_data.csv" file.*

*REFERENCES*

Buehler, S., Oberson, A., Rao, I. M., Friesen, D. K. and Frossard, E.: Sequential Phosphorus Extraction of a 33P-Labeled Oxisol under Contrasting Agricultural Systems, Soil Sci. Soc. Am. J., 66(3), 868–877, doi:10.2136/sssaj2002.8680, 2002.

Frossard, E., Brossard, M., Hedley, M. J. and Metherell, A.: Reactions controlling the cycling of P in soils, in Phosphorus in the Global Environment: Transfers, Cycles, and Management, edited by H. Tiessen, pp. 107–138, John Wiley & Sons, Ltd., 1995.

Frossard, E., Condron, L. M., Oberson, A., Sinaj, S. and Fardeau, J. C.: Processes Governing Phosphorus Availability in Temperate Soils, J. Environ. Qual., 29(1), doi:10.2134/jeq2000.00472425002900010003x, 2000.

Guo, F., Yost, R. S., Hue, N. V., Evensen, C. I. and Silva, J. A.: Changes in phosphorus fractions in soils under intensive plant growth, Soil Sci. Soc. Am. J., 64(5), 1681–1689, doi:10.2136/sssaj2000.6451681x, 2000.

Helfenstein, J., Tamburini, F., von Sperber, C., Massey, M., Pistocchi, C., Chadwick, O., Vitousek, P., Kretzschmar, R. and Frossard, E.: Combining spectroscopic and isotopic techniques

gives a dynamic view of phosphorus cycling in soil, Nat. Commun., 9, doi:10.1038/s41467-018-05731-2, 2018.

Syers, J. K., Johnston, A. E. and Curtin, D.: Efficiency of soil and fertilizer phosphorus use: Reconciling changing concepts of soil phosphorus behaviour with agronomic information, FAO, Rome. [online] Available from: http://www.fao.org/docrep/010/a1595e/a1595e00.htm, 2008.

Tamburini, F., Pfahler, V., Bünemann, E. K., Guelland, K., Bernasconi, S. M. and Frossard, E.: Oxygen isotopes unravel the role of microorganisms in phosphate cycling in soils, Environ. Sci. Technol., 46(11), 5956–5962, doi:10.1021/es300311h, 2012.

---

## Author Response (AR2)

**Author response to 2nd Reviewer**

"I am generally happy at the authors' responses and revisions. Following are a few topics that are open to discuss, which are largely of my personal opinion."

*Thank you very much for supporting the publication of our manuscript and for your final constructive remarks. Below, we address your comments point by point and describe our changes to the final manuscript version.*

"(1) I agree that practically solution P can exchange with HCl-P in not only acidic soils but also neutral and alkaline soils. However, the way the authors showed in Fig. 1a is misleading. Fig. 1a suggests that solution P is totally exchangeable with HCl-P, which is not true. I would suggest that the arrow from resin P to HCl-P should be shorter than that from HCl-P to resin P, and explain the settings: exchangeable but the net flow is from HCl-P to resin P (rephrase it)."

*The conceptual scheme reported in Figure 1 is taken and modified after Fardeau (1993). It is indeed referred to exchanges between HCl-P and the solution without net change in the pool size. This kind of situation is experimentally observed on the short and medium term, see for example Pistocchi et al 2018.*

*We have now reported the original reference, which was missing (sorry for that) to better clarify the context to which this conceptual model is referred. We added a sentence (p.10 l. 2-3) to acknowledge the difference with exchanges at pedogenic time scales in which the HCl-P pool is progressively depleted.*

"(2) The calculation of MRT of NaOH-P and HCl-P still puzzle me. If possible, MRT of NaOH-P (or HCl-P) should be calculated as the MRT of NaOH fraction (or HCl-P fraction), not the sum of resin P and NaOH fraction (or HCl fraction). If calculated in the current way, the estimated MRT would be not directly comparable with the MRT used in processes-based models (i.e., MRT of each fraction). If it is impossible to calculate the MRT of individual fraction, the authors may note the problem of comparing with processes-based models."

*We're sorry that this was still not clear enough in the manuscript. We know explained it even more so that it should make sense to readers not familiar with the approach (see p. 6):*

*"In sequential extractions, P pools are sequentially removed from the soil, and this has to be accounted in the calculation of MRT. MRT of resin P and labile P was calculated using using Eq. (3) and plugging in resin-P or labile-P pool sizes for Ppool. However, for NaOH-P and HCl-P the previously removed P pools have to be formally "added back". Hence, for NaOH-P and HCl-P the Ppool was set equal to the sum of labile-P and NaOH-P or labile-P, NaOH-P, and HCl-P respectively (Fig. 1b). Not accounting for the sequential nature of these pools and using NaOH-P or HCl-P for Ppool directly in Eq. (3) would considerably underestimate MRT."*

(3) The authors should be cautious about the statement "current LSM largely overestimate MRTs of P in inorganic P pools" and their comparison with LSM (Table 3). Because LSM typically

simulate soil P dynamics in field stations, while the estimates here are under laboratory conditions. I am not surprise to see results in Table 3. Table 3 does not necessarily means "current LSM largely overestimate MRTs of P in inorganic P pools". So, the authors should make readers aware of the differences under laboratory vs. field conditions.

*We don't think that the large differences in MRT are because our data is from laboratory conditions. As shown in the discussion, our data fits quite well with data available from longer incubation experiments (several months) (see 11., l. 1-10) and field experiments (see p. 11, l. 15-23). We agree with the reviewer that in the field, MRT may differ, but not by orders of magnitude. We acknowledge the limitation that our data is from lab experiments (p. 12, l. 7) and also conclude the paper by saying that more data is needed, also from field experiments (p. 14, l. 1).*

*Hence we did not see it necessary to make additional changes.*

"(4) In acknowledgements, "Enquing" should be "Enqing"."

*We have corrected. Sorry for the mistake!*

*REFERENCES*

[revised manuscript text omitted]